

# Measurement experiences with FluxSet digital D/I station

László Hegymegi[a], János Szöllősy[b], Csaba Hegymegi[a], and Ádám Domján[a]

[a]MinGeo Ltd., Budapest, Hungary
[b]Individual contractor

*Correspondence to:* László Hegymegi (hegymegi@mingeo.com)

**Abstract.** Geomagnetic observatories use classical theodolites equipped with single axis flux-gate magnetometers known as declination inclination magnetometers (DIM) to determine absolute values of declination and inclination angles. This instrument and the measurement method is very reliable but needs a lot of handwork and experience. The authors developed and built a non-magnetic theodolite which gives all measurement data in digital form. Use of this instrument significantly decreases possibility of observation errors and minimizes handwork. The new instrument is presented in this paper together with first measurement results in comparison to the classical DIM.

## 1 Introduction

Regular absolute measurement of the geomagnetic field using classical DIM is required to ensure the stability and calibration of geomagnetic observatory data (St-Louis et al., 2012). A digital non-magnetic DIM known as the Declination/Inclination Digital Station 1 (DS-1) was designed and built to improve the quality and reliability of the manual process.

This instrument measures declination and inclination angles of the geomagnetic field vector with digital angle encoders as well as the magnetic field measured by the attached single axis FluxSet magnetometer. Data are transmitted via radio to a central processor unit. This unit also receives time stamp information from a built-in GPS receiver. All these data are stored in the central unit for later processing.

The central processor unit has a serial port which connects to a local scalar absolute magnetometer. This solution can be useful in case of field measurements when the distance to the nearest observatory is too great to provide adequate correlation.

Data visualization is made by portable devices such as laptop, tablet or e-book reader using WiFi communication with, the 802.11 AC standard. The connected unit is also used to setup configuration parameters. At the end of measurements all data can be exported to an xls or csv formatted table.

All the instrumentation run on built-in batteries for at least three hours and can be charged using 12 V DC or 230 V AC. The data visualization display shows the power state of the batteries for every component of the DS-1.

Comparative measurements were made between the DS-1 and the Nagycenk Observatory's absolute instrument on 19 December 2016. The final appraisal shows that the DS-1 is suitable for absolute control measurements, and it is more convenient, user friendly and effective than the traditional DIM. More comparison measurements will be performed for much longer periods with weekly intervals, to check the operation stability and precision of the DS-1.




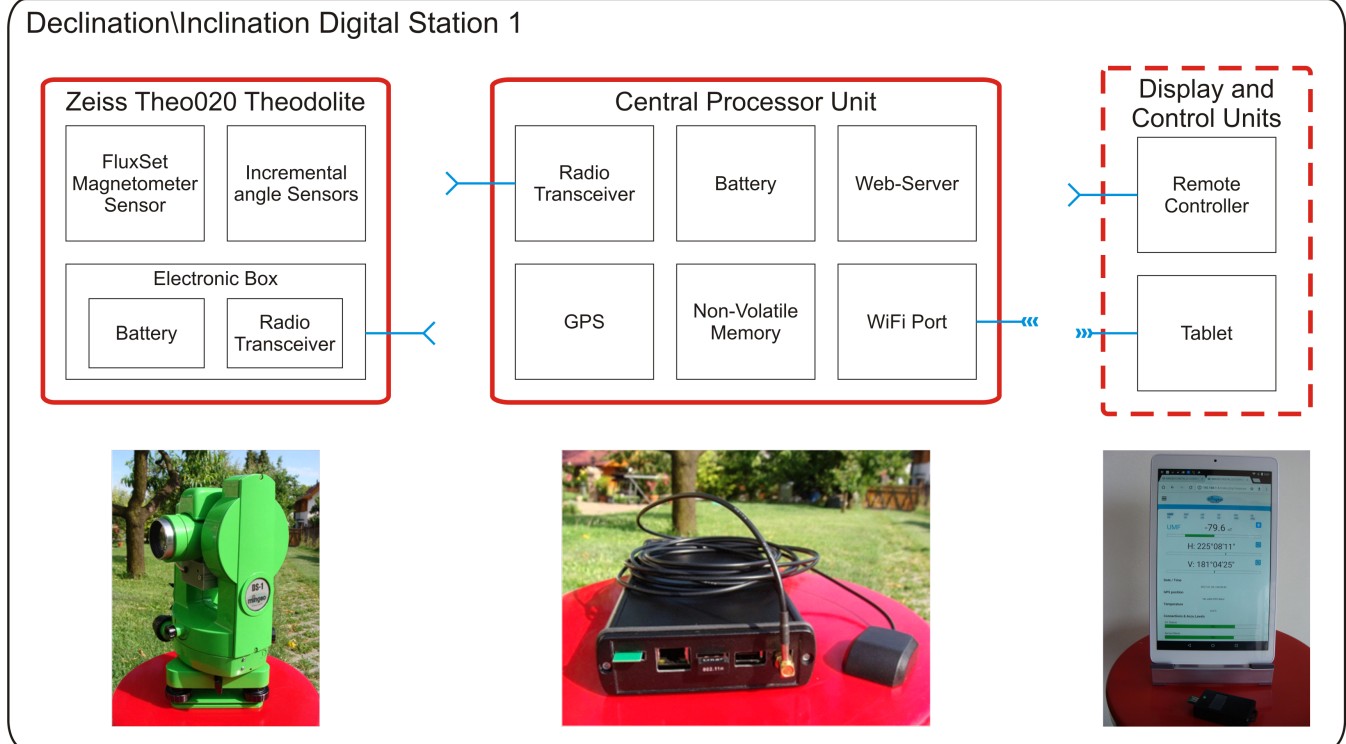

**Figure 1.** Declination/Inclination Digital Station 1 (DS-1)

## 2 Hardware construction

Hardware components of the first model of the DS-1 equipment is shown on **Figure 1**. It has three main parts, the theodolite with a single axis magnetometer, the central processor and the display unit.

The most important part is the modified Zeiss THEO 020 theodolite mechanics, with built-in digital encoders and a FluxSet
5   magnetometer mounted on its telescope. All steel and all inner optical parts of the original theodolite are removed and original graduated circles are replaced with the new glass circles of the digital angle encoder. The vertical encoder chip is fixed on the pendulum of the theodolite to eliminate reading differences caused by small levelling errors. The magnetic sensor was placed on the top of the telescope. The magnetometer has a resolution of 0.1 nT and the dynamic range of operation is +/- 3000 nT. All the electronics of the angle encoders and magnetometer are built in to the electronics housing and fixed on the bottom
10   of the telescope of the instrument. It also contains the battery with a charge controller and a radio unit to provide cable-less connection to the central processor and to the remote controller. Owing to this design the small ( < 1 nT ) magnetic effect of the printed circuit board in the electronic box is cancelled by measurements taken in four positions, because it rotates with the telescope.





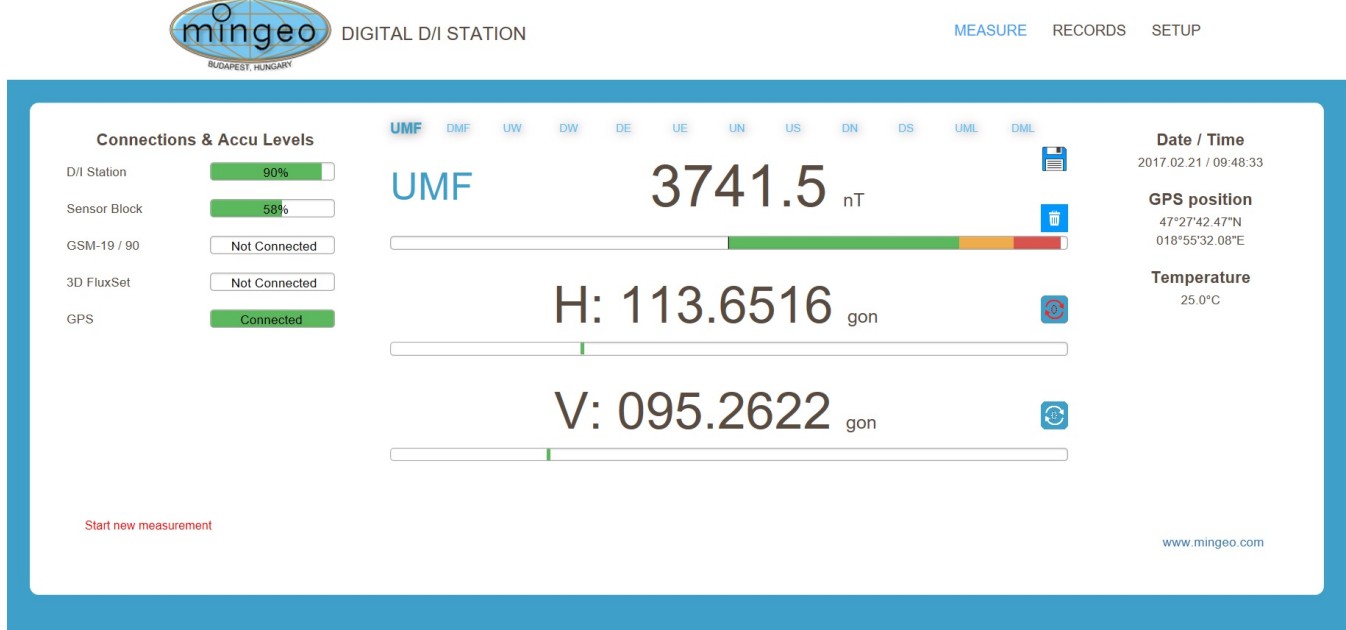

**Figure 2.** The measurement tab of the GUI web-page. On the left side there are connection and battery life informations. On the right side there are date, time, GPS and temperature data. In the middle of the tab is shown the current magnetic sensor reading with its corresponding horizontal and vertical angle values. In top-middle the predefined measurement sequence can be seen, which begins with the first Azimuth reading with telescope up position (UMF) and emphasis on the ongoing position.

The central processor unit is responsible for the WiFi network deployment, data storage, radio communication between the hardware components, and provides a web-server having a GUI web-page. The central processor unit receives measured magnetic and angle values and measurement commands sent by the observer via remote controller. It has a built-in GPS receiver to supply time stamp for measurement data and geographical coordinates which can be useful for field use. It has a connection

5   possibility for absolute scalar magnetometer if these data are not available from another external source. The unit has its own battery ensuring operation for about six hours. It has some steel parts therefore it must be placed a few meters away from the absolute pillar. The central processor unit electronics does not have built-in display only LEDs indicating regular operation and charging process. The display function is performed by a WiFi capable tablet. A large screen tablet is included in the package of DS-1 equipment. However the user can use his any WiFi capable device web browser functionality. After DS-1

10   installation the operator can follow the absolute magnetic measurement watching the GUI web-page on the tablet and saving the measurements via the remote controller.





## 3  Measurement experience and results

The measurement procedure with DS-1 is the same as with DIM. On the display screen in addition to magnetometer output, horizontal (*H*) and vertical (*V*) angle values with some other information is shown on the web-service display. There is also a reminder of the following sensor and telescope position in the measurement protocol (**Figure 2**). Pushing a knob on the remote

5 controller the actual data is stored in the memory of DS-1.

In order to judge quality of absolute values provided by DS-1 equipment a one day long instrument comparison test was carried out at Nagycenk Observatory (NCK)on 19 December 2016. The reference instrument was NCK's main absolute instrument. The measurements were taken alternately by two operators.

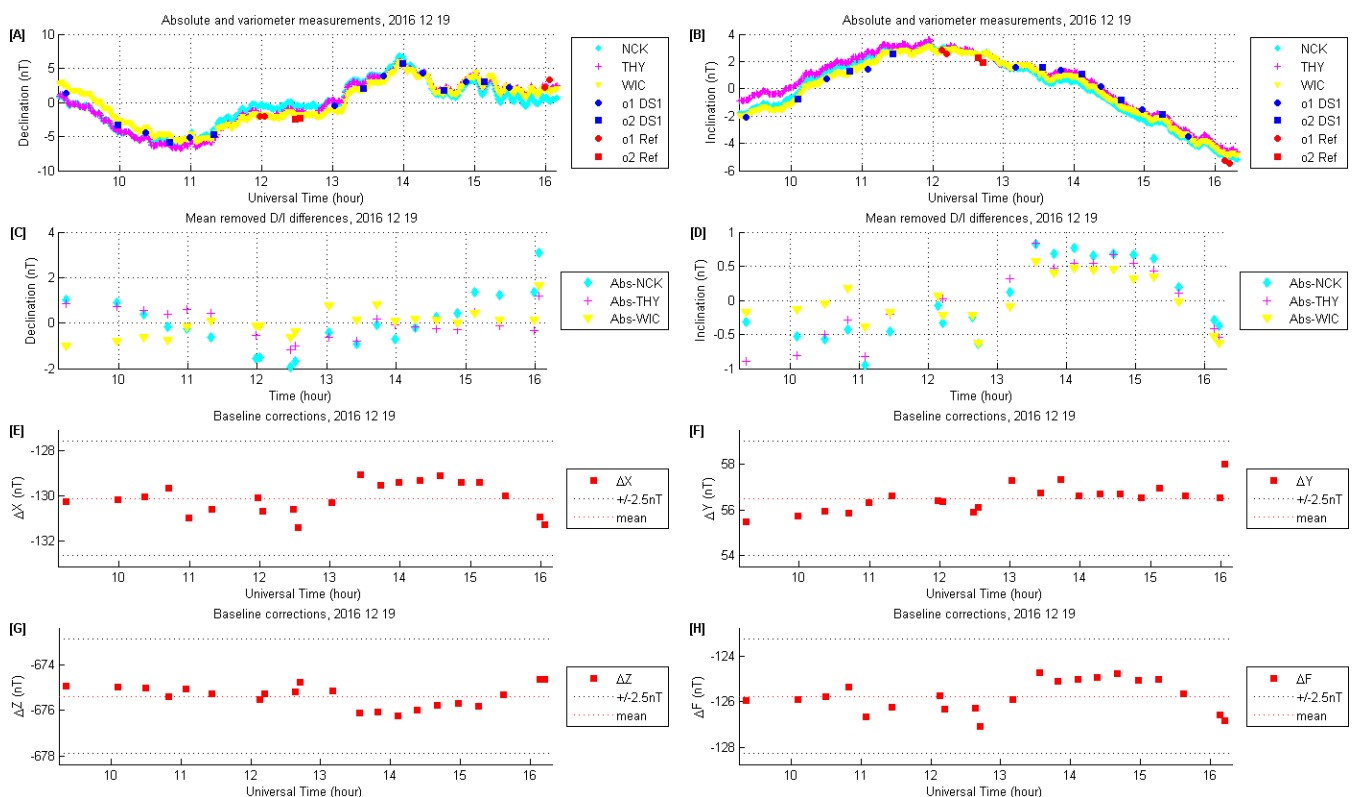

**Figure 3. A**-**B**: The mean-removed Declination and Inclination values in nT unit. The circle and square mark is referring to observer 1 and 2, respectively. The red color corresponds to the absolute measurements with the reference instrument, while the blue color to DS-1 equipment. The diamond-cyan, cross-magenta and the yellow-triangle markers designate the calculated declination and inclination from the one-minute variometer data of the NCK, THY and WIC observatories, respectively. **C**-**D**: Difference of the absolute and averaged *D*, *I* angles. **E**-**H**: The baseline corrections calculated from absolute data for the related components. The correction values accomplished the adopted confidence limit in the INTERMAGNET specification (Definitive baseline data accuracy: +/-5nT). (St-Louis et al., 2012)


During the observations the geomagnetic activity was quiet. The difference of maximum and minimum value was within 21nT at *X* and *Y* horizontal components. (Clarke et al., 2013) The observers followed the duplicate, symmetrical scheme (*MDIIDM*, *M* stands for the Azimuth (mira), *D* is the declination angle, *I* is the inclination angle) to produce a set of readings with the reference instrument, in order to easily identify any error as this is the usual measurement routine at NCK. Typical

operators require about 20-25 minutes to complete one measurement series. On the 2016 IAGA Observatory Workshop in Dourbes the simplified scheme (*MDIM*) was used. The use of different schemas was complicated analysis, but it did not affect the outcome of the results as the NCK main instrument's absolute data could be converted into (*MDIM*) format.

During the comparison test on 19 December 2016 we completed 15 series with the DS-1 and 3 series with the reference instrument using the simplified and the duplicate scheme, respectively. Absolute measurements with the reference instrument

were at the middle and the end of the day. Results are shown on **Figure 3A** and **3B**, respectively. For comparison *D*, *I* angles are also calculated from the one-minute average of variometer data of NCK, THY and WIC observatories. (Zhang et al., 2016) It is clearly seen that the absolute *D*, *I* angles follow the calculated one-minute variometer data during the day.

On **Figure 3C** and **3D** we show the difference of measured *D*, *I* angles and average *D*, *I* values computed from variometer data of NCK, THY and WIC observatories for the period of measurements. Statistical results are shown on **Table 1**. The

standard deviation of the differences is below 1.5 nT for every components.

**Table 1.** The statistical parameters of the differences between absolute and variometer Declination/Inclination angles

| Declination differences | | | |
|---|---|---|---|
| | Abs-NCK (nT) | Abs-THY (nT) | Abs-WIC (nT) |
| Mean | -73.66 | -52.68 | 66.16 |
| St Dev | 1.23 | 0.64 | 0.61 |
| Variance | 1.52 | 0.41 | 0.37 |
| Max-Min | 5.02 | 2.33 | 2.68 |

| Inclination differences | | | |
|---|---|---|---|
| | Abs-NCK (nT) | Abs-THY (nT) | Abs-WIC (nT) |
| Mean | 392.91 | -207.46 | 79.31 |
| St Dev | 0.56 | 0.57 | 0.37 |
| Variance | 0.32 | 0.33 | 0.14 |
| Max-Min | 1.78 | 1.74 | 1.19 |

**Table 2.** Statistics of baseline corrections at all and individual instruments

| Statistics of all absolute measurements | | | | |
|---|---|---|---|---|
| | $\Delta X$ (nT) | $\Delta Y$ (nT) | $\Delta Z$ (nT) | $\Delta F$ (nT) |
| Mean | -130.11 | 56.52 | -675.37 | -125.76 |
| St Dev | 0.71 | 0.59 | 0.49 | 0.71 |
| Variance | 0.50 | 0.35 | 0.24 | 0.51 |
| Max-Min | 2.32 | 2.57 | 1.64 | 2.37 |

| Statistics of DS-1 equipment abs. measurements | | | | |
|---|---|---|---|---|
| | $\Delta X$ (nT) | $\Delta Y$ (nT) | $\Delta Z$ (nT) | $\Delta F$ (nT) |
| Mean | -129.82 | 56.50 | -675.52 | -125.47 |
| St Dev | 0.57 | 0.55 | 0.46 | 0.58 |
| Variance | 0.32 | 0.30 | 0.21 | 0.34 |
| Max-Min | 1.88 | 1.88 | 1.34 | 1.93 |

| Statistics of Ref. instrument abs. measurements | | | | |
|---|---|---|---|---|
| | $\Delta X$ (nT) | $\Delta Y$ (nT) | $\Delta Z$ (nT) | $\Delta F$ (nT) |
| Mean | -130.83 | 56.56 | -675.00 | -126.47 |
| St Dev | 0.49 | 0.76 | 0.38 | 0.47 |
| Variance | 0.24 | 0.57 | 0.15 | 0.22 |
| Max-Min | 1.34 | 2.13 | 0.91 | 1.36 |



We calculated $\Delta X$, $\Delta Y$, $\Delta Z$, $\Delta F$ corrections which have to be added to $X$, $Y$, $Z$ and $F$ baseline components, according to 5.1 section of (Jankowski and Sucksdorff, 1996, p. 86) to serve as a secondary quality check. For these calculations we used the total field readings of the continuously operating proton precession scalar magnetometer of the NCK observatory. The corresponding plots of the corrections shown on the **Figure 3E-H**. Statistical properties are summarized in **Table 2**. All

5 corrections show low standard deviation . They are inside within +/-2.5 nT interval, which suggests that it could be a good source to baseline fitting. Furthermore, these standard deviations demonstrate that the DS-1 is suitable to supply high quality data for IMOs.

Certainly one days data is not enough for an accurate baseline determination, but it is a good sign that the data exhibits low scatter characteristics. Our next mission is to perform at least a week a long observation with DS-1 equipment to see data

quality of this instrument and check whether it meets the INTERMAGNET standard. (St-Louis et al., 2012), (Newitt et al., 1996)

## 4   Conclusions

The DS-1 equipment is good enough to determine the absolute values of $D$, $I$ elements of the Earth's magnetic field vector based on statistics of the one full day absolute data measured by the Digital Station 1. The confidence limit of calculated

baseline corrections are verified to meet the INTERMAGNET standards and specifications. (St-Louis et al., 2012), (Newitt et al., 1996)

Use of DS-1 equipment is very comfortable especially at polar or equatorial regions, where a classical instrument must use a 90° eyepiece to read the vertical circle. Another benefit of the DS-1 equipment is that most if not all transcription errors are eliminated. This is especially useful in the field or when the operator uses the "residual" method.

The DS-1 equipment can be employed for the repeat station surveys. A local scalar magnetometer can be connected to the serial port of the central processor unit giving the possibility to produce absolute elements of the geomagnetic field vector.

## 5   Future Work

We are planning to carry out a one year experiment in cooperation with NCK observatory's stuff to compare DS-1 equipment to the NCK observatory's main instrument in production of annual baseline.

Due to the convenient design and simple data acquisition with DS-1 it opens the possibility to testing and evaluating new and more sophisticated absolute measurement routines beyond the "null" or the "residual" method. This equipment enables the observer to obtain much more data in a shorty time. A new method is currently under development such at the Niemegk observatory (Brunke and Matzka, 2016), (Brunke and Matzka, 2017) which can be tested easily with DS-1. Common experiments are planned.



*Acknowledgements.* We are thankful to the stuff of Geodetic and Geophysical Institute of Hungary (Judit Szendrői, Viktor Wesztergom) who provided expertise that greatly assisted the research, and the possibility to compare our instrument with theirs and the access of all kind of NCK observatory data.





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
