# Peer review of "Measurement experiences with FluxSet digital D/I station"

_Geoscientific Instrumentation, Methods and Data Systems, 2017_

## Referee Comment (RC1) · H.-U. Auster (Referee) · 13 Mar 2017

H.-U. Auster (Referee)

uli.auster@tu-bs.de

Reading the angles of a non magnetic theodolite digitally could be a great improvement for the DI-Flux absolute measurement. It is a great development and I recommend to publish the paper if the following minor comments are considered.

The main question, how much the digital reading influences the measurement quality, should be pointed out more clearly. The only quality check is a set of absolute measurements at one (magnetically quiet) day, which tells us, that the DS-1 works. The error bars (even if three are presented, St-Dev, Variance and Max-Min) can not be related to specific sources. $\Delta F$ is based on external data only and not relevant for the quality of the DS-1 system. Thus a more dedicated check of the potential risk of introducing the digital reading shall be added.

[Figure]

In detail: (1) Please add a more specific test checking the precision of the angular readings. One option could be that you measure the field component for arbitrary angular settings (within the fluxgate range of +/-3000nT) and compare it with the nominal field component calculated for the angles. The field vector should be known in an observatory at any time, and the setup parameter (orientation vs. azimuth mark, fluxgate offset, fluxgate orientation) are known from at least one absolute measurement with the DS-1 system.

(2) Reducing the influence of the nearby electronics (including battery and RF unit) to a level below 1nT is a remarkable achievement. For more confidence, results of a magnetic characterization of the electronics should be added which should includes remanent and induced (electronics might contain soft magnetic material) contributions as well as the effect of perm and deperm.

(3) Using the single component fluxgate sensor for absolute measurements by the residual method with larger deviations from zero crossing (as indicated in Figure 2) the author should mention how he eliminates non linearity errors and scale values uncertainties because only offset and alignment errors are vanished by the DI-Flux measurement procedure.

Last but not least I want to point out again the importance of this development because absolute measurements could become more robust and observer skills less important. • With the DS-1 only leveling of the theodolite and focusing of the azimuth mark still need human skills. • The opportunity of reading angles digitally could improve the method using a three axes fluxgate on top of the telescope of the non magnetic theodolite significantly. • With a synchronous reading of the fluxgate data, external variometer and scalar data, the dependency on field activity should be reduced / vanished.

---

## Referee Comment (RC2) · A. Gonsette (Referee) · 19 Apr 2017

**Review of Hegymegi et al., "Measurement experiences with FluxSet digital D/I station"**

The paper focuses on a digital version of a non-magnetic theodolite based on a Zeiss 020 instrument. Such device is therefore suitable for magnetic observatories as well as for in-field measurements. I recommend to publish the paper if authors consider the following comments:

Chapter 2 is a little light. In particular, authors do not give any information about the modified theodolite performances. The angular accuracy according to a specific norm (e.g. ISO 17-123) is not given. The pendulum has been modified in order to host the V-circle reading head. What is the effect on the tilt compensation? Does the temperature have an impact on the angle reading?

The FluxSet sensor remains somehow exotic for the magnetic observatory community. Authors have mentioned it in the title and should therefore spend little energy for a (short) description. What is the difference between a FluxSet and a more conventional Fluxgate?

Page 2: "Owing to this design…because it rotates with the telescope." This is not evident for the reader. Author should include an experiment or a reference: (Gilbert D and Rasson JL, (1998). Effect on DIflux Measuring Accuracy due to a Magnet located on it, Proceedings of the VIIth Workshop on Geomagnetic Observatory Instruments, Data Acquisition and Processing, Scientific Technical Report STR98/21, pp168-171, GeoForschungsZentrum Potsdam.)

Moreover, it is true if and only if the magnetization remains constant during the whole set of 4 measurements. Is it the case?

Page 3 line 6: What is the minimum distance between DIM and CPU? What is the CPU magnetic signature at 1meter when it is switched on? What is the tablet impact?

Chapter 3: The authors should give some details about the measurement procedure (p4-l1). Residual or zero method? What sequence?

Also, neither a variometer description nor the reference instrument description are present. Are DS-1 and ref-DIM on the same pillar?

A robust comparison between two DIM is made by computing a variometer baselines for more than 1 day. Then baselines are compared. Such elementary validation test is not present in the paper. Fig 3 E-H are just a few points (3x2 points for reference instrument) plot. Authors promise a one-year comparison as future work. Maybe they could already provide 1 or 2 weeks comparison.

Page 5: Why the duplicated MDIIDM scheme? Is 20-25 min for the traditional DIM or for the DS-1? Could authors detail the handwork gain compared to traditional instruments?

---

## Author Comment (AC1) · 30 May 2017

**Author's final response**

**Answers to comments of H.-U. Auster:**

**Comment 1:**

Please add a more specific test checking the precision of the angular readings. One option could be that you measure the field component for arbitrary angular settings (within the fluxgate range of +/-3000nT) and compare it with the nominal field component calculated for the angles. The field vector should be known in an observatory at any time, and the setup parameter (orientation vs. azimuth mark, fluxgate offset, fluxgate orientation) are known from at least one absolute measurement with the DS-1 system.

*Author's response:*

Not exactly the proposed check was carried out but absolute measurement using the inverse method proposed by (Brunke and Matzka, 2017), where the readings were taken in different telescope position and results were similar to the results got using the conventional method. This suggests that there is no problem with the angular precision. Anyway a detailed angular reading precision test is in preparation.

*Author's changes in manuscript:*

At page 7 line 17 the "We also plan to check the instrument according to the ISO 17123-3 standard. (ISO 17123-3:2001)" sentence was inserted.

**Comment 2:**

Reducing the influence of the nearby electronics (including battery and RF unit) to a level below 1nT is a remarkable achievement. For more confidence, results of a magnetic characterization of the electronics should be added which should includes remanent and induced (electronics might contain soft magnetic material) contributions as well as the effect of perm and deperm.

*Author's response:*

The electronics box have remanent magnetization which is less than 20 nT in total from 1 cm distance. However the electronics is mounted on the telescope, and therefore its effect is canceled out. We have not tested yet the induced magnetism of the electronic box. Our one day test suggests its effect could be small if there is any. Perm and deperm procedure was not carried out.

*Author's changes in manuscript:*

At page 3 line 5 the "Owing to this design the  magnetic effect of the of the printed circuit board and battery in the electronic box is cancelled by measurements taken in four positions, because it rotates with the telescope  (Gilbert and Rasson, 1998). We suppose that the box has not any induced magnetism however it was not tested yet." sentences were augmented and revised.

Comment 3:

Using the single component fluxgate sensor for absolute measurements by the residual method with larger deviations from zero crossing (as indicated in Figure 2) the author should mention how he eliminates non linearity errors and scale values uncertainties because only offset and alignment errors are vanished by the DI-Flux measurement procedure.

*Author's response:*

The single component FluxSet sensor is linear inside its operating range. On (new) Figure 2 we can see that the FluxSet magnetometer is linear inside the +/-3000 nT operation range and the maximum difference of observations is 3.2nT from the calculated line, while the PPM reading variations were maximum 2 nT during the measurements. Every DS-1 instrument's scale factor is determined and stored in its non volatile memory. This value has to be checked from time to time.

*Author's changes in manuscript:*

A (new) Figure 2 was added, with its caption text. At page 3 line 2 the "In this range the magnetometer is linear as it is shown on Figure 2." sentence was inserted.

**Answers to comments of A. Gonsette:**

Comment 1:

Chapter 2 is a little light. In particular, authors do not give any information about the modified theodolite performances. The angular accuracy according to a specific norm (e.g. ISO 17-123) is not given.

*Author's response:*

The ISO 17123 specific norm is not yet available for us. It is under purchase. When we will have, then the angular accuracy measurement will be performed accordingly.

*Author's changes in manuscript:*

At page 7 line 17 the "We also plan to check the instrument according to the ISO 17123-3 standard. (ISO 17123-3:2001)" sentence was inserted.

Comment 2:

The pendulum has been modified in order to host the V-circle reading head. What is the effect on the tilt compensation?

*Author's response:*

The tilt compensation was checked and the modification had no any effect on it.

*Author's changes in manuscript:*

At page 2 line 7 the "This modification has not any influence on the smooth operation of the pendulum, which was proved experimentally." sentence was inserted.

Comment 3:

Does the temperature have an impact on the angle reading?

*Author's response:*

There is in a built in temperature sensor to monitor temperature changes but we did not experienced any temperature effect on the angle reading.

*Author's changes in manuscript:*

There was no change added.

Comment 4:

The FluxSet sensor remains somehow exotic for the magnetic observatory community. Authors have mentioned it in the title and should therefore spend little energy for a (short) description. What is the difference between a FluxSet and a more conventional Fluxgate?

*Author's response:*

Chapter 2 was revised and a short description of the FluxSet magnetometer was added.

*Author's changes in manuscript:*

At page 2 line 9 the " The DS-1 instrument is built with FluxSet magnetometer. Its operating principle is similar to the pulse-position type fluxgate magnetometer. Practical advantage of these

magnetometers that their signal can be easily converted intoa binary signal and the measurement of magnetic fieldis reduced toa high precision time measurement through the displacement of the magnetisation curve produced by the external field. The magnetometer measures the axial magnetic field at the probe. The transverse sensitivity is negligible. The probe of the device is a small size coil around arounda high permeability (one mm wide) amorphous metal strip." sentences were inserted.

Comment 5:

Page 2: "Owing to this design…because it rotates with the telescope." This is not evident for the reader. Author should include an experiment or a reference: (Gilbert D and Rasson JL, (1998). Effect on DIflux Measuring Accuracy due to a Magnet located on it, Proceedings of the VIIth Workshop on Geomagnetic Observatory Instruments, Data Acquisition and Processing, Scientific Technical Report STR98/21, pp168-171, GeoForschungsZentrum Potsdam.)

*Author's response:*

We put the corresponding reference into that paragraph.

*Author's changes in manuscript:*

At page 3 line 5 the "Owing to this design the  magnetic effect of the of the printed circuit board and battery in the electronic box is cancelled by measurements taken in four positions, because it rotates with the telescope  (Gilbert and Rasson, 1998)." sentence was revised.

Comment 6:

Moreover, it is true if and only if the magnetization remains constant during the whole set of 4 measurements. Is it the case?

*Author's response:*

It is supposed that the remanent magnetization of the electronics box during the measurements remains constant as it can change mainly with changing temperature but this is nearly constant during the short period of one absolute measurement.

*Author's changes in manuscript:*

There was no change added.

Comment 7:

Page 3 line 6: What is the minimum distance between DIM and CPU? What is the CPU magnetic signature at 1meter when it is switched on? What is the tablet impact?

*Author's response:*

The minimum distance between the electronic unit, tablet and the FluxSet sensor is 2m. From that distance the effect of both units is below 0.5 nT.

*Author's changes in manuscript:*

At page 3 line 14 the "It has some steel parts therefore  its magnetic effect from 1 and 2 meter is 3.0 and 0.3 nT, respectively. It must be placed  at least 2 meters away from the absolute pillar." sentences were corrected and augmented. At page 3 line 17 the "A large screen tablet is included in the package of DS-1 equipment, the magnetic effect of which is 3.5 and 0.5 nT from 1 and 2 meter. Instead of this tablet, the user can use his any WiFi capable device  which has web browser." sentences were corrected and augmented.

Comment 8:

Chapter 3: The authors should give some details about the measurement procedure (p4-l1). Residual or zero method? What sequence?

*Author's response:*

All absolute magnetic measurement were measured by the null method. The utilized sequences were mentioned in page 4 line 11.

*Author's changes in manuscript:*

At page 4 line 3 the " Basically should be used like a classical DIM." sentence was added.

Comment 9:

Also, neither a variometer description nor the reference instrument description are present. Are DS-1 and ref-DIM on the same pillar?

*Author's response:*

The registration of geomagnetic field variations is carried out by two sets of triaxial fluxgate magnetometers, namely by the ARGOS and the DRXX systems at NCK observatory. The NCK observatory main absolute instrument is a Zeiss THEO 010B theodolite equipped with DTU single axis fluxgate magnetometer model G. The measurements were taken the same pillar.

*Author's changes in manuscript:*

At page 4 line 8 the "The reference instrument was NCK's main absolute instrument, a ZEISS THE 010B equipped with DMI G type fluxgate magnetometer." sentence was augmented. At page 5 line 6 the "The instruments were installed on the same absolute pillar of the NCK observatory" sentence was added. At page 5 line 9 "At NCK the registration of geomagnetic field variations is carried out by two sets of triaxial fluxgate magnetometers, namely by the ARGOS and the DRXX systems. (Ádám et al., 2009) The other two observatories' variometer data were downloaded from the INTERMAGNET website." sentences were added.

Comment 10:

A robust comparison between two DIM is made by computing a variometer baselines for more than 1 day. Then baselines are compared. Such elementary validation test is not present in the paper. Fig 3 E-H are just a few points (3x2 points for reference instrument) plot. Authors promise a one-year comparison as future work. Maybe they could already provide 1 or 2 weeks comparison.

*Author's response:*

The long term comparison test measurements are in progress. There is no 2 week comparison test yet.

*Author's changes in manuscript:*

There was no change added.

Comment 11:

Page 5: Why the duplicated MDIIDM scheme?

*Author's response:*

The NCK stuff use the MDIIDM scheme. With their instrument we measured with their standard in order to they can put the raw data into their absolute processing program. With the DS-1 we measured with IAGA 2016 Dourbes format.

*Author's changes in manuscript:*

At page 4 line 11 the " At the NCK measurement procedures the duplicate, symmetrical scheme (MDIIDM, M stands for the  azimuth (mark), D is the declination angle, I is the inclination angle)is used to produce a set of readings with the reference instrument, in order to easily identify any error . Typical operators require about 20-25 minutes to complete  this schema with both instrument. However with the DS-1 the simplified scheme (MDIM) was used like at the 2016

IAGA Observatory Workshop in Dourbes. The use of different schemas  made the analysis more complicated, but it  had no effect on the results." sentences were revised.

Comment 12:

Is 20-25 min for the traditional DIM or for the DS-1?

*Author's response:*

The duplicate symmetrical MDIIDM schema and the simplified MDIM schema take to 20-25 and 15-18 minutes to complete them also DS-1 and the reference instrument, respectively.

*Author's changes in manuscript:*

At page 4 line 15 the "Typical operators require about 20-25 minutes to complete  this schema  with both instrument." sentence was added.

Comment 13:

Could authors detail the handwork gain compared to traditional instruments?

*Author's response:*

In the DS-1 the telescope is used for mark sightings adjustment only. There is no need to read the circle readings, because you see the telescope positions on the tablet. You do not need to go around the pillar during the whole measurement. You do not need to handle the cable of the magnetometer. You do not need to write on any data just to push a button on the remote controller. After the measurement do not need to digitize the measurement protocol, since it is in digital form already.

*Author's changes in manuscript:*

There is no change added.

**Author's self corrections:**

In the (new) Figure 4, table 2 and the body text the ΔF was replaced with ΔH, since this is the appropriate notation for the horizontal baseline corrections.

---

## Author Comment (AC3) · 30 May 2017

**Author's final response**

**Answers to comments of H.-U. Auster:**

Comment 1:

Please add a more specific test checking the precision of the angular readings. One option could be that you measure the field component for arbitrary angular settings (within the fluxgate range of +/-3000nT) and compare it with the nominal field component calculated for the angles. The field vector should be known in an observatory at any time, and the setup parameter (orientation vs. azimuth mark, fluxgate offset, fluxgate orientation) are known from at least one absolute measurement with the DS-1 system.

*Author's response:*

Not exactly the proposed check was carried out but absolute measurement using the inverse method proposed by (Brunke and Matzka, 2017), where the readings were taken in different telescope position and results were similar to the results got using the conventional method. This suggests that there is no problem with the angular precision. Anyway a detailed angular reading precision test is in preparation.

*Author's changes in manuscript:*

At page 7 line 17 the "We also plan to check the instrument according to the ISO 17123-3 standard. (ISO 17123-3:2001)" sentence was inserted.

Comment 2:

Reducing the influence of the nearby electronics (including battery and RF unit) to a level below 1nT is a remarkable achievement. For more confidence, results of a magnetic characterization of the electronics should be added which should includes remanent and induced (electronics might contain soft magnetic material) contributions as well as the effect of perm and deperm.

*Author's response:*

The electronics box have remanent magnetization which is less than 20 nT in total from 1 cm distance. However the electronics is mounted on the telescope, and therefore its effect is canceled out. We have not tested yet the induced magnetism of the electronic box. Our one day test suggests its effect could be small if there is any. Perm and deperm procedure was not carried out.

*Author's changes in manuscript:*

At page 3 line 5 the "Owing to this design the  magnetic effect of the of the printed circuit board and battery in the electronic box is cancelled by measurements taken in four positions, because it rotates with the telescope  (Gilbert and Rasson, 1998). We suppose that the box has not any induced magnetism however it was not tested yet." sentences were augmented and revised.

Comment 3:

Using the single component fluxgate sensor for absolute measurements by the residual method with larger deviations from zero crossing (as indicated in Figure 2) the author should mention how he eliminates non linearity errors and scale values uncertainties because only offset and alignment errors are vanished by the DI-Flux measurement procedure.

*Author's response:*

The single component FluxSet sensor is linear inside its operating range. On (new) Figure 2 we can see that the FluxSet magnetometer is linear inside the +/-3000 nT operation range and the maximum difference of observations is 3.2nT from the calculated line, while the PPM reading variations were maximum 2 nT during the measurements. Every DS-1 instrument's scale factor is determined and stored in its non volatile memory. This value has to be checked from time to time.

*Author's changes in manuscript:*

A (new) Figure 2 was added, with its caption text. At page 3 line 2 the "In this range the magnetometer is linear as it is shown on Figure 2." sentence was inserted.

**Answers to comments of A. Gonsette:**

Comment 1:

Chapter 2 is a little light. In particular, authors do not give any information about the modified theodolite performances. The angular accuracy according to a specific norm (e.g. ISO 17-123) is not given.

*Author's response:*

The ISO 17123 specific norm is not yet available for us. It is under purchase. When we will have, then the angular accuracy measurement will be performed accordingly.

*Author's changes in manuscript:*

At page 7 line 17 the "We also plan to check the instrument according to the ISO 17123-3 standard. (ISO 17123-3:2001)" sentence was inserted.

Comment 2:

The pendulum has been modified in order to host the V-circle reading head. What is the effect on the tilt compensation?

*Author's response:*

The tilt compensation was checked and the modification had no any effect on it.

*Author's changes in manuscript:*

At page 2 line 7 the "This modification has not any influence on the smooth operation of the pendulum, which was proved experimentally." sentence was inserted.

Comment 3:

Does the temperature have an impact on the angle reading?

*Author's response:*

There is in a built in temperature sensor to monitor temperature changes but we did not experienced any temperature effect on the angle reading.

*Author's changes in manuscript:*

There was no change added.

Comment 4:

The FluxSet sensor remains somehow exotic for the magnetic observatory community. Authors have mentioned it in the title and should therefore spend little energy for a (short) description. What is the difference between a FluxSet and a more conventional Fluxgate?

*Author's response:*

Chapter 2 was revised and a short description of the FluxSet magnetometer was added.

*Author's changes in manuscript:*

At page 2 line 9 the " The DS-1 instrument is built with FluxSet magnetometer. Its operating principle is similar to the pulse-position type fluxgate magnetometer. Practical advantage of these

magnetometers that their signal can be easily converted intoa binary signal and the measurement of magnetic fieldis reduced toa high precision time measurement through the displacement of the magnetisation curve produced by the external field. The magnetometer measures the axial magnetic field at the probe. The transverse sensitivity is negligible. The probe of the device is a small size coil around arounda high permeability (one mm wide) amorphous metal strip." sentences were inserted.

**Comment 5:**

Page 2: "Owing to this design…because it rotates with the telescope." This is not evident for the reader. Author should include an experiment or a reference: (Gilbert D and Rasson JL, (1998). Effect on DIflux Measuring Accuracy due to a Magnet located on it, Proceedings of the VIIth Workshop on Geomagnetic Observatory Instruments, Data Acquisition and Processing, Scientific Technical Report STR98/21, pp168-171, GeoForschungsZentrum Potsdam.)

*Author's response:*

We put the corresponding reference into that paragraph.

*Author's changes in manuscript:*

At page 3 line 5 the "Owing to this design the  magnetic effect of the of the printed circuit board and battery in the electronic box is cancelled by measurements taken in four positions, because it rotates with the telescope  (Gilbert and Rasson, 1998)." sentence was revised.

**Comment 6:**

Moreover, it is true if and only if the magnetization remains constant during the whole set of 4 measurements. Is it the case?

*Author's response:*

It is supposed that the remanent magnetization of the electronics box during the measurements remains constant as it can change mainly with changing temperature but this is nearly constant during the short period of one absolute measurement.

*Author's changes in manuscript:*

There was no change added.

**Comment 7:**

Page 3 line 6: What is the minimum distance between DIM and CPU? What is the CPU magnetic signature at 1meter when it is switched on? What is the tablet impact?

*Author's response:*

The minimum distance between the electronic unit, tablet and the FluxSet sensor is 2m. From that distance the effect of both units is below 0.5 nT.

*Author's changes in manuscript:*

At page 3 line 14 the "It has some steel parts therefore  its magnetic effect from 1 and 2 meter is 3.0 and 0.3 nT, respectively. It must be placed  at least 2 meters away from the absolute pillar." sentences were corrected and augmented. At page 3 line 17 the "A large screen tablet is included in the package of DS-1 equipment, the magnetic effect of which is 3.5 and 0.5 nT from 1 and 2 meter. Instead of this tablet, the user can use his any WiFi capable device  which has web browser." sentences were corrected and augmented.

**Comment 8:**

Chapter 3: The authors should give some details about the measurement procedure (p4-l1). Residual or zero method? What sequence?

*Author's response:*

All absolute magnetic measurement were measured by the null method. The utilized sequences were mentioned in page 4 line 11.

*Author's changes in manuscript:*

At page 4 line 3 the " Basically should be used like a classical DIM." sentence was added.

**Comment 9:**

Also, neither a variometer description nor the reference instrument description are present. Are DS-1 and ref-DIM on the same pillar?

*Author's response:*

The registration of geomagnetic field variations is carried out by two sets of triaxial fluxgate magnetometers, namely by the ARGOS and the DRXX systems at NCK observatory. The NCK observatory main absolute instrument is a Zeiss THEO 010B theodolite equipped with DTU single axis fluxgate magnetometer model G. The measurements were taken the same pillar.

*Author's changes in manuscript:*

At page 4 line 8 the "The reference instrument was NCK's main absolute instrument, a ZEISS THE 010B equipped with DMI G type fluxgate magnetometer." sentence was augmented. At page 5 line 6 the "The instruments were installed on the same absolute pillar of the NCK observatory" sentence was added. At page 5 line 9 "At NCK the registration of geomagnetic field variations is carried out by two sets of triaxial fluxgate magnetometers, namely by the ARGOS and the DRXX systems. (Ádám et al., 2009) The other two observatories' variometer data were downloaded from the INTERMAGNET website." sentences were added.

Comment 10:

A robust comparison between two DIM is made by computing a variometer baselines for more than 1 day. Then baselines are compared. Such elementary validation test is not present in the paper. Fig 3 E-H are just a few points (3x2 points for reference instrument) plot. Authors promise a one-year comparison as future work. Maybe they could already provide 1 or 2 weeks comparison.

*Author's response:*

The long term comparison test measurements are in progress. There is no 2 week comparison test yet.

*Author's changes in manuscript:*

There was no change added.

Comment 11:

Page 5: Why the duplicated MDIIDM scheme?

*Author's response:*

The NCK stuff use the MDIIDM scheme. With their instrument we measured with their standard in order to they can put the raw data into their absolute processing program. With the DS-1 we measured with IAGA 2016 Dourbes format.

*Author's changes in manuscript:*

At page 4 line 11 the " At the NCK measurement procedures the duplicate, symmetrical scheme (MDIIDM, M stands for the  azimuth (mark), D is the declination angle, I is the inclination angle)is used to produce a set of readings with the reference instrument, in order to easily identify any error . Typical operators require about 20-25 minutes to complete  this schema with both instrument. However with the DS-1 the simplified scheme (MDIM) was used like at the 2016

IAGA Observatory Workshop in Dourbes. The use of different schemas  made the analysis more complicated, but it  had no effect on the results." sentences were revised.

Comment 12:

Is 20-25 min for the traditional DIM or for the DS-1?

*Author's response:*

The duplicate symmetrical MDIIDM schema and the simplified MDIM schema take to 20-25 and 15-18 minutes to complete them also DS-1 and the reference instrument, respectively.

*Author's changes in manuscript:*

At page 4 line 15 the "Typical operators require about 20-25 minutes to complete  this schema  with both instrument." sentence was added.

Comment 13:

Could authors detail the handwork gain compared to traditional instruments?

*Author's response:*

In the DS-1 the telescope is used for mark sightings adjustment only. There is no need to read the circle readings, because you see the telescope positions on the tablet. You do not need to go around the pillar during the whole measurement. You do not need to handle the cable of the magnetometer. You do not need to write on any data just to push a button on the remote controller. After the measurement do not need to digitize the measurement protocol, since it is in digital form already.

*Author's changes in manuscript:*

There is no change added.

**Author's self corrections:**

In the (new) Figure 4, table 2 and the body text the ΔF was replaced with ΔH, since this is the appropriate notation for the horizontal baseline corrections.

**Measurement experiences with FluxSet digital D/I station**

László Hegymegi[a], János Szöllősy[b], Csaba Hegymegi[a], and Ádám Domján[a]

[a]MinGeo Ltd., Budapest, Hungary
[b]Individual contractor

*Correspondence to:* László Hegymegi (hegymegi@mingeo.com)

**Abstract.** Geomagnetic observatories use classical theodolites equipped with single axis flux-gate magnetometers known as declination inclination magnetometers (DIM) to determine absolute values of declination and inclination angles. This instrument and the measurement method is very reliable but needs a lot of handwork and experience. The authors developed and built a non-magnetic theodolite which gives all measurement data in digital form. Use of this instrument significantly decreases possibility of observation errors and minimizes handwork. The new instrument is presented in this paper together with first measurement results in comparison to the classical DIM.

**1 Introduction**

Regular absolute measurement of the geomagnetic field using classical DIM is required to ensure the stability and calibration of geomagnetic observatory data (St-Louis et al., 2012). A digital non-magnetic DIM known as the Declination/Inclination Digital Station 1 (DS-1) was designed and built to improve the quality and reliability of the manual process.

This instrument measures declination and inclination angles of the geomagnetic field vector with digital angle encoders as well as the magnetic field measured by the attached single axis FluxSet magnetometer. Data are transmitted via radio to a central processor unit. This unit also receives time stamp information from a built-in GPS receiver. All these data are stored in the central unit for later processing.

The central processor unit has a serial port which connects to a local scalar absolute magnetometer. This solution can be useful in case of field measurements when the distance to the nearest observatory is too great to provide adequate correlation.

Data visualization is made by portable devices such as laptop, tablet or e-book reader using WiFi communication with, the 802.11 AC standard. The connected unit is also used to setup configuration parameters. At the end of measurements all data can be exported to an xls or csv formatted table.

All the instrumentation run on built-in batteries for at least three hours and can be charged using 12 V DC or 230 V AC. The data visualization display shows the power state of the batteries for every component of the DS-1.

Comparative measurements were made between the DS-1 and the Nagycenk Observatory's absolute instrument on 19 December 2016. The final appraisal shows that the DS-1 is suitable for absolute control measurements, and it is more convenient, user friendly and effective than the traditional DIM. More comparison measurements will be performed for much longer periods with weekly intervals, to check the operation stability and precision of the DS-1.

[Figure]

**Figure 1.** Declination/Inclination Digital Station 1 (DS-1)

**2  Hardware construction**

Hardware components of the first model of the DS-1 equipment is shown on **Figure 1**. It has three main parts, the theodolite with a single axis magnetometer, the central processor and the display unit.

The most important part is the modified Zeiss THEO 020 theodolite mechanics, with built-in digital encoders and a FluxSet magnetometer mounted on its telescope. All steel and all inner optical parts of the original theodolite are removed and original graduated circles are replaced with the new glass circles of the digital angle encoder. The vertical encoder chip is fixed on the pendulum of the theodolite to eliminate reading differences caused by small levelling errors.  This modification has not any influence on the smooth operation of the pendulum, which was proved experimentally.

The DS-1 instrument is built with FluxSet magnetometer. Its operating principle is similar to the pulse-position type fluxgate magnetometer. Practical advantage of these magnetometers that their signal can be easily converted into a binary signal and the measurement of magnetic field is reduced to a high precision time measurement through the displacement of the magnetisation curve produced by the external field. The magnetometer measures the axial magnetic field at the probe. The transverse sensitivity is negligible. The probe of the device is a small size coil around around a high permeability (one mm wide) amorphous metal strip. (Vértessy et al., 2000)

[Figure]

**Figure 2.** FluxSet magnetometer is linear inside the +/-3000 nT range and the observations deviate with 3.2nT from the calculated line (left), while the scalar (PPM) readings changed maximum 2 nT during the 10 minutes taken linearity check test (right).

The magnetic sensor was placed on the top of the telescope. The magnetometer has a resolution of 0.1 nT and the dynamic range of operation is +/- 3000 nT. In this range the magnetometer is linear as it is shown on **Figure 2**.

All the electronics of the angle encoders and magnetometer are built in to the electronics housing and fixed on the bottom of the telescope of the instrument. It also contains the battery with a charge controller and a radio unit to provide cable-less
5   connection to the central processor and to the remote controller. Owing to this design the  magnetic effect of the  printed circuit board and battery in the electronic box is cancelled by measurements taken in four positions, because it rotates with the telescope  (Gilbert and Rasson, 1998). We suppose that the box has not any induced magnetism however it was not tested yet.

The central processor unit is responsible for the WiFi network deployment, data storage, radio communication between
10  the hardware components, and provides a web-server having a GUI web-page. The central processor unit receives measured magnetic and angle values and measurement commands sent by the observer via remote controller. It has a built-in GPS receiver to supply time stamp for measurement data and geographical coordinates which can be useful for field use. It has a connection possibility for absolute scalar magnetometer if these data are not available from another external source. The unit has its own battery ensuring operation for about six hours. It has some steel parts therefore  its magnetic effect from 1 and 2 meter is 3.0
15  and 0.3 nT, respectively. It must be placed  at least 2 meters away from the absolute pillar. The central processor unit electronics does not have built-in display only LEDs indicating regular operation and charging process.

The display function is performed by a  tablet equipped with WiFi. A large screen tablet is included in the package of DS-1 equipment, the magnetic effect of which is 3.5 and 0.5 nT from 1 and 2 meter. Instead of this tablet, the user can use his any WiFi capable device  which has web browser. After DS-1
20  installation the operator can follow the absolute magnetic measurement watching the GUI web-page on the tablet and saving the measurements via the remote controller.

[Figure]

**Figure 3.** The measurement tab of the GUI web-page. On the left side there are connection and battery life informations. On the right side there are date, time, GPS and temperature data. In the middle of the tab is shown the current magnetic sensor reading with its corresponding horizontal and vertical angle values. In top-middle the predefined measurement sequence can be seen, which begins with the first  azimuth reading with telescope up position (UMF) and emphasis on the ongoing position.

**3  Measurement experience and results**

The measurement procedure with DS-1 is the same as with DIM. Basically should be used like a classical DIM. On the display screen in addition to magnetometer output, horizontal (*H*) and vertical (*V*) angle values with some other information is shown on the web-service display. There is also a reminder of the following sensor and telescope position in the measurement protocol

5    (**Figure 3**). Pushing a knob on the remote controller the actual data is stored in the memory of DS-1.

In order to judge quality of absolute values provided by DS-1 equipment a one day long instrument comparison test was carried out at Nagycenk Observatory (NCK)on 19 December 2016. The reference instrument was NCK's main absolute instrument, a ZEISS THE 010B equipped with DMI G type fluxgate magnetometer. The measurements were taken alternately by two operators.

10    During the observations the geomagnetic activity was quiet. The difference of maximum and minimum value was within 21nT at *X* and *Y* horizontal components. (Clarke et al., 2013)  At the NCK measurement procedures the duplicate, symmetrical scheme (*MDIIDM*, *M* stands for the azimuth (mark), *D* is the declination angle, *I* is the inclination angle) is used to produce a set of readings with the reference instrument, in order to easily identify any error. Typical operators require about 20-25 minutes to complete

series.  this schema with both instrument. However with the DS-1 the simplified scheme (*MDIM*) was used like at the 2016 IAGA Observatory Workshop in Dourbes. The use of different schemas  made the analysis more complicated, but it   had no effect on the results.

5     During the comparison test on 19 December 2016 we completed 15 series with the DS-1 and 3 series with the reference instrument using the simplified and the duplicate scheme, respectively. The instruments were installed on the same absolute pillar of the NCK observatory. Absolute measurements with the reference instrument were at the middle and the end of the day. Results are shown on **Figure 4A** and **4B**, respectively. For comparison *D*, *I* angles are also calculated from the one-minute average of variometer data of NCK, THY and WIC observatories. (Zhang et al., 2016) At NCK the registration of geomagnetic
10   field variations is carried out by two sets of triaxial fluxgate magnetometers, namely by the ARGOS and the DRXX systems.

[Figure]

**Figure 4. A**-**B**: The mean-removed Declination and Inclination values in nT unit. The circle and square mark is referring to observer 1 and 2, respectively. The red color corresponds to the absolute measurements with the reference instrument, while the blue color to DS-1 equipment. The diamond-cyan, cross-magenta and the yellow-triangle markers designate the calculated declination and inclination from the one-minute variometer data of the NCK, THY and WIC observatories, respectively. **C**-**D**: Difference of the absolute and averaged *D*, *I* angles. **E**-**H**: The baseline corrections calculated from absolute data for the related components. The correction values accomplished the adopted confidence limit in the INTERMAGNET specification (Definitive baseline data accuracy: +/-5nT). (St-Louis et al., 2012)

**Table 1.** The statistical parameters of the differences between absolute and variometer Declination/Inclination angles

| Declination differences | Abs-NCK (nT) | Abs-THY (nT) | Abs-WIC (nT) |
|---|---|---|---|
| Mean | -73.66 | -52.68 | 66.16 |
| St Dev | 1.23 | 0.64 | 0.61 |
| Variance | 1.52 | 0.41 | 0.37 |
| Max-Min | 5.02 | 2.33 | 2.68 |

| Inclination differences | Abs-NCK (nT) | Abs-THY (nT) | Abs-WIC (nT) |
|---|---|---|---|
| Mean | 392.91 | -207.46 | 79.31 |
| St Dev | 0.56 | 0.57 | 0.37 |
| Variance | 0.32 | 0.33 | 0.14 |
| Max-Min | 1.78 | 1.74 | 1.19 |

**Table 2.** Statistics of baseline corrections at all and individual instruments

| Statistics of all absolute measurements | $\Delta X$ (nT) | $\Delta Y$ (nT) | $\Delta Z$ (nT) |  $\Delta H$ (nT) |
|---|---|---|---|---|
| Mean | -130.11 | 56.52 | -675.37 | -125.76 |
| St Dev | 0.71 | 0.59 | 0.49 | 0.71 |
| Variance | 0.50 | 0.35 | 0.24 | 0.51 |
| Max-Min | 2.32 | 2.57 | 1.64 | 2.37 |

| Statistics of DS-1 equipment abs. measurements | $\Delta X$ (nT) | $\Delta Y$ (nT) | $\Delta Z$ (nT) |  $\Delta H$ (nT) |
|---|---|---|---|---|
| Mean | -129.82 | 56.50 | -675.52 | -125.47 |
| St Dev | 0.57 | 0.55 | 0.46 | 0.58 |
| Variance | 0.32 | 0.30 | 0.21 | 0.34 |
| Max-Min | 1.88 | 1.88 | 1.34 | 1.93 |

| Statistics of Ref. instrument abs. measurements | $\Delta X$ (nT) | $\Delta Y$ (nT) | $\Delta Z$ (nT) |  $\Delta H$ (nT) |
|---|---|---|---|---|
| Mean | -130.83 | 56.56 | -675.00 | -126.47 |
| St Dev | 0.49 | 0.76 | 0.38 | 0.47 |
| Variance | 0.24 | 0.57 | 0.15 | 0.22 |
| Max-Min | 1.34 | 2.13 | 0.91 | 1.36 |

(Ádám et al., 2009) The other two observatories' variometer data were downloaded from the INTERMAGNET website. It is clearly seen that the absolute $D$, $I$ angles follow the calculated one-minute variometer data during the day.

[revised manuscript text omitted]

Gilbert, D. and Rasson, J. L.: Effect on DIflux Measuring Accuracy due to a Magnet located on it, Scientific Technical Report STR98/21,, 1998.

ISO 17123-3:2001: Optics and optical instruments – Field procedures for testing geodetic and surveying instruments – Part 3: Theodolites, Standard, International Organization for Standardization, Geneva, CH, https://www.iso.org/obp/ui/#iso:std:iso:17123:-3:ed-1:v1:en, 2001.

15 Jankowski, J. and Sucksdorff, C.: IAGA Guide for Magnetic Measurements and Observatory Practice, International Association of Geomagnetism and Aeronomy, https://books.google.hu/books?id=UCM8PQAACAAJ, 1996.

Newitt, L. R., Barton, C. E., and Bitterly, J.: Guide for Magnetic Repeat Station, International Association of Geomagnetism and Aeronomy, http://ftp.bcmt.fr/pdf/IAGA-Guide-Repeat-Stations.pdf, 1996.

St-Louis, B. J., Sauter, E. A., and Coles, R. L.: InterMagnet Technical Reference Manual, version 4.6, InterMagnet, http://www.intermagnet.
20 org/publication-software/technicalsoft-eng.php, 2012.

Vértessy, G., Gasparics, A., and Szöllősy, J.: High sensitivity magnetic field sensor, Sensors and Actuators A: Physical, 85, 202 – 208, doi:https://doi.org/10.1016/S0924-4247(00)00368-X, http://www.sciencedirect.com/science/article/pii/S092442470000368X, 2000.

Zhang, S., Fu, C., He, Y., Yang, D., Li, Q., Zhao, X., and Wang, J.: Quality Control of Observation Data by the Geomagnetic Network of China, http://doi.org/10.5334/dsj-2016-015, 2016.